# Cellular Geometry Sensing at Different Length Scales and its Implications for Scaffold Design

**DOI:** 10.3390/ma13040963

**Published:** 2020-02-21

**Authors:** Maike Werner, Nicholas A. Kurniawan, Carlijn V. C. Bouten

**Affiliations:** 1Soft Tissue Engineering and Mechanobiology, Department of Biomedical Engineering, Eindhoven University of Technology, 5612 AP Eindhoven, The Netherlands; m.werner@tue.nl (M.W.); c.v.c.bouten@tue.nl (C.V.C.B.); 2Institute for Complex Molecular Systems (ICMS), Eindhoven University of Technology, 5612 AZ Eindhoven, The Netherlands

**Keywords:** tissue engineering, biomaterial, scaffold, contact guidance, substrate curvature, geometry sensing, mechanobiology, cell migration, cytoskeleton, cell nucleus

## Abstract

Geometrical cues provided by the intrinsic architecture of tissues and implanted biomaterials have a high relevance in controlling cellular behavior. Knowledge of how cells sense and subsequently respond to complex geometrical cues of various sizes and origins is needed to understand the role of the architecture of the extracellular environment as a cell-instructive parameter. This is of particular interest in the field of tissue engineering, where the success of scaffold-guided tissue regeneration largely depends on the formation of new tissue in a native-like organization in order to ensure proper tissue function. A well-considered internal scaffold design (i.e., the inner architecture of the porous structure) can largely contribute to the desired cell and tissue organization. Advances in scaffold production techniques for tissue engineering purposes in the last years have provided the possibility to accurately create scaffolds with defined macroscale external and microscale internal architectures. Using the knowledge of how cells sense geometrical cues of different size ranges can drive the rational design of scaffolds that control cellular and tissue architecture. This concise review addresses the recently gained knowledge of the sensory mechanisms of cells towards geometrical cues of different sizes (from the nanometer to millimeter scale) and points out how this insight can contribute to informed architectural scaffold designs.

## 1. Introduction

In vivo, tissues have specific geometrical properties that contribute to their physiological functions. For instance, bone tissue has a hierarchical structure that is critical for its mechanical properties [1,2,3]. Tendons and ligaments have highly aligned collagen fibers that endow these tissues with the ability to withstand mechanical loading and thereby prevent tissue damage [4]. Another example is healthy cardiovascular tissues that are generally organized in an anisotropic manner in order to withstand the high directional strains and shear stresses present in these tissues and to perform their functions effectively [5,6,7,8,9]. Within these tissues there is a dynamic interplay between the cells and the extracellular matrix (ECM) [10,11]. Cellular orientation can be affected by the tissue geometry. In turn, these cells contribute to the architectural organization of the extracellular matrix by rearranging the ECM and producing new matrix proteins [12]. Knowledge of how the geometry of the extracellular environment influences cell behavior, and vice versa, is crucial to gain a better understanding of healthy tissue development, as well as the microenvironmental situations underlying changes in cellular behavior in a disease [13,14]. Furthermore, it opens an attractive avenue of using substrate geometry as an instructive factor to direct cellular organization towards a native situation. This is of particular relevance in the field of tissue engineering, where biomaterial scaffolds can be specifically designed to guide tissue regeneration [15,16].

In tissue engineering, a three-dimensional (3D) biodegradable material can be utilized as a temporary template to promote and guide the neo-tissue formation in a pre-determined manner [17]. Especially in in-situ tissue engineering, in which case a cell-free scaffold is implanted and cells need to actively infiltrate the scaffold, it is of interest that the scaffold properties can guide cells into a desirable organization [18]. Ultimately, the infiltrating cells should form a tissue with specific organization that grants proper tissue functionality. Conventional scaffold production techniques allowed only rough tuning of the geometrical properties of the scaffold [19]. Scaffold fabrication techniques have rapidly advanced and have started to allow increasingly detailed control over the geometry of the product [20,21,22,23,24,25,26]. Additive manufacturing techniques can produce 3D structures in a layer-by-layer fashion using a 3D computer-aided design (CAD) [27]. This has allowed the design and the production of complex geometries with high accuracy. The external geometry of the scaffold can be designed in a patient-specific manner, based on medical imaging data (e.g., from MRI or CT scans) [17]. Moreover, this approach also allows detailed control of the internal geometry, even using mathematically defined functions [19,28,29]. Reproducible control of the internal scaffold architecture plays an essential role in employing the scaffold internal design to guide cell infiltration and tissue organization in the desired direction. A large variety of different pore structures from the nanometer to mesoscopic scale can be combined in scaffolds to create complex multi-scale hierarchical scaffold geometries [20,30,31,32,33]. Some examples of these scaffold designs are presented in Figure 1.

Even though extensive control of scaffold design is currently possible, rational design of scaffold geometries that directs cell organization to achieve specific tissue structures is not yet a commonly taken approach currently and remains a challenge, as it relies on prior knowledge on how cells sense and respond towards basic geometries [29,34]. Attempts to tune several scaffold structural parameters to direct cellular responses have been summarized in recent reviews [35,36]. Moreover, the scaffold’s internal architecture is characterized by multiple interdependent parameters (such as porosity, pore size, strut size, strut geometry, etc.), making it difficult to attribute an observed cellular reaction to one particular parameter. A further challenge is that the architectural properties also have an effect on the mechanical and degradation properties of the scaffold, which in turn can influence the cellular behavior and tissue structure as well [34,37].

Minimal model systems with highly defined and reproducible characteristics have been used to identify and understand the cellular response towards specific cues [38,39]. Over the last decades, studies employing two-dimensional (2D) or pseudo-3D two-and-a-half-dimensional (2.5D) relief structures with highly controlled geometrical cues have demonstrated that cells respond to a wide variety of geometrical cues in the form of ridges, lines, pillars, pits or other shapes in sizes ranging from nanometer to millimeter scale (reviewed by [38,40]). These fabricated geometrical cues are designed to resemble the geometrical cues that cells are subjected to in vivo, in the form of both native tissue and biomimetic scaffolds, which are known to influence various cellular responses. For example, neuron growth is strongly guided by the directions of scaffold fibers, leading to various approaches to induce nerve repair by tuning the properties of biomaterials and scaffolds [41,42]. Another example is in the case of (micro)vascular tissue engineering, where multiple cell types need to organize in different configurations (e.g., endothelial cells in a concave tubular construct, wrapped around by pericytes) to ensure function within the microvasculature [43,44]. Moreover, increasing attention has been focused toward the question of how cells actually perceive these geometrical cues of the extracellular environment, in addition to the downstream effect of the cue on the cellular behavior. This growing mechanistic insight in geometry sensing is essential to understand how cells will respond to increasingly complex geometric environments. Scaffold designers could use the knowledge of how cells sense and respond to different geometrical cues to make informed, smart designs where the cellular orientation and subsequent tissue organization can be predicted and steered in the desired direction.

The numerous downstream effects of geometry on cell behavior and the effect of geometry on the mechanical function of the scaffold have been extensively discussed in recent reviews (see, for example, [40,47,48] and [34,37], respectively). Here, we concisely review the recently gained mechanistic insights on how cells sense geometrical cues of different sizes and origins. We will first briefly summarize the known general principles of how cells sense mechanical and physical properties of their environment. Next, we focus on the mechanisms that cells employ to sense geometrical guidance cues smaller than cell size, which are of relevance for the design of fibrous scaffolds with fiber diameters ranging from nanometer to a few micrometers. Recent, unexpected discoveries that cells can also sense geometrical cues larger than cell size are highlighted. In particular, we focus on the sensing of substrate curvature, an important parameter in 3D tissue engineering scaffolds produced by additive manufacturing. Finally, we propose several promising research directions to further advance our knowledge on the cellular geometry sensing and response in complex environments and describe how this could contribute to smart scaffold design.

## 2. Cellular Geometry Sensing

Cellular adhesions to the substrate, the cytoskeleton, and the nucleoskeleton all play important roles in how adherent cells sense geometrical cues of their substrate. When a cell adheres, spreads, and polarizes on a substrate, actin polymerizes into fibrils, forming filamentous actin (F-actin). F-actin pushes the plasma membrane forward, creating a leading edge protrusion. Integrins in the plasma membrane function as anchorage points to the substrate. Integrins are intermembrane proteins with an extracellular tail and an intracellular tail [49]. Activated integrins attach with their extracellular tail to ECM molecules such as fibronectin, vitronectin, collagen, and laminin, while the intracellular component of the protein is attached to F-actin through adapter proteins (such as talin). This connection forms a nascent adhesion of the cell with the substrate [50]. F-actin can further organize into higher order structures called stress fibers. These stress fibers contain crosslinked F-actin and non-muscle myosin motor proteins. This composition enables stress fibers to contract [51,52]. Following F-actin polymerization at the leading edge, the F-actin networks are moved backwards towards the cell center. This F-actin retrograde flow, resulting from F-actin polymerization and myosin contractility, creates a force directed towards the center of the cell on the nascent adhesions [53,54,55]. The traction force pulls on the adhesions and thereby drives the maturation of focal adhesions (FAs) by triggering a cascade of events where integrins cluster together and large number of adapter proteins (such as focal adhesion kinase (FAK), paxillin, and vinculin) are recruited and subsequently activated to strengthen the bond of the ECM to the cytoskeleton of the cell [50,56]. Thus, FA represents the protein complexes that physically link the intracellular cytoskeleton to the extracellular matrix. FAs function as a so-called molecular clutch. When a force applied on the adhesion is below a certain threshold, the FA disintegrates, however a force above the threshold further strengthens the adhesion [57,58,59].

Actin fibers are not only connected to the FAs, but they can also be physically linked to the nucleoskeleton of the nucleus at the nuclear membrane via protein linkers. Nesprin proteins at the outer nuclear membrane are linked to SUN proteins, located in the inner nuclear membrane, which connect to the nuclear lamina and chromatin at the inside of the nucleus [60,61]. Chromatin contains the genetic information of the cell, the DNA, which needs to be protected. The viscoelastic properties of the nucleus ensure that it can be protected against deformations. A-type lamins (lamin A and C) in the nuclear lamina are proteins that can be considered as viscous components that organize in a condensed state when force is applied to the nucleus and thereby function as a shock-absorber [62]. B-type lamins, on the other hand, have elastic properties that restore the nuclear shape [63]. The LInker of the Nucleoskeleton with the Cytoskeleton (LINC complex) shows resemblance to how the cytoskeleton is linked to the ECM via FA complexes at the cellular membrane. In fact, through this continuous connection, the nuclear interior is mechanically linked to the cellular exterior via the LINC complexes, cytoskeletal fibers, and FA complexes (Figure 2A) [61].

Cells can employ this direct physical interconnection between intracellular components and ECM to detect mechanical properties from their environment, such as ECM stiffness and topography. Cells can transmit traction forces (e.g., generated by myosin contraction or actin-retrograde flow) through the adhesions on the ECM. The forces acting on the cell–matrix interactions depend on the ECM properties and the cellular contractility [58]. This force can give rise to mechanosensing via a number of molecular mechanosensors, proteins that respond to the force application (e.g., via conformation changes or changed binding activity) [57]. This can induce a series of different signaling events that transduce the mechanical signals in the cell (termed mechanotransduction) and result in cellular changes in order for the cell to adapt to the extracellular environment (termed mechanoresponse) [64]. 

In the following sections, we will focus our attention on how cells employ the connection of the nucleus through the cytoskeleton and the FAs to sense geometrical cues from the extracellular environment. In particular, we will discuss in detail how focal adhesion placement, growth, and stability seem to be the determining factors in how cells sense anisotropic physical cues at size ranges smaller than the size of the cell, while F-actin integrity and nucleus placement appear to determine the cellular orientation response towards geometries larger than cell size.

## 3. Adhesion Position, Maturation, and Stability Form the Basis of Contact Guidance Sensing

Contact guidance is a phenomenon that is defined by cellular orientation in the direction of anisotropic cues on the substrate [70]. Cues for contact guidance can, for instance, be found in the form of aligned collagen fibrils [14,71] or fibrous electrospun scaffolds often used in tissue engineering [5,72,73]. Minimal model systems such as printed protein lines, substrates with nano- or microtopographies or single fibers have been utilized to systematically investigate the cellular sensing mechanism of such cues. The force-dependent maturation and stabilization of FAs that was described above has been found to be a key mechanism by which cells perceive contact guidance cues. The size, shape, and orientation of the features on the substrate can for instance regulate adhesion maturation by controlling the available area for adhesion growth [74,75,76]. Doyle et al. compared the adhesion dynamics in fibroblasts when cultured on 2D substrates and on a 1D-like single 1.5–2 µm-wide line. In contrast to on 2D surfaces, all FAs on 1D-like substrates matured. The adhesions on the lines remained stable six times longer than on a 2D surface. Furthermore, cells showed higher protrusion rates on 1D lines compared to 2D surfaces [75]. This indicates that directional constriction of the available attachment area induces FA maturation in a predefined orientation. Recently, Ray et al. showed with various human breast and pancreatic carcinoma cell lines on aligned ridged substrates (ridge and groove width 800 nm) that the topography of the substrate can constrict FA growth, only allowing the FA to grow in the direction where attachment molecules are present. This leads to directional maturation and orientation of FAs and is accompanied by an aligned F-actin organization, leading to anisotropic traction forces. These directed traction forces, in turn, bias the cell polarization and migration direction on aligned substrates [74].

It is important to note that local anisotropic forces can also give rise to a biased FA growth. A single collagen fiber has stiffness in the MPa range in the longitudinal direction, however the stiffness is much lower in the perpendicular direction [77]. It was demonstrated in fibroblasts that FAs and leading edge protrusions were oriented in the longitudinal direction of the collagen fibers in an aligned 3D collagen gel. In this situation, FA growth was not necessarily limited by the substrate feature size or orientation, but rather the higher rigidity of the fiber along the longitudinal axis promoted the cell to form stable FAs in this direction. The local orientation-dependent fiber rigidity has an effect on FA stability, which subsequently provides an anchor for further protrusion in the direction of the longitudinal axis of the fiber [77].

FA maturation and stability is force-rate-dependent and this rate can be influenced by internal factors such as myosin II contractility [57], which in turn can be influenced by external factors (e.g., substrate feature size and shape, and substrate rigidity). Myosin inhibition by blebbistatin resulted in a reduced protrusion rate at the leading edge in cells on a 1D adhesive line. This suggests that myosin II contractility is vital for increased FA stability, which in turn is associated with leading edge protrusion [75]. Interestingly, Kubow et al. recently reported that contact guidance is independent of myosin II activity. They observed that both control and myosin II inhibited cells formed adhesions along aligned polycaprolactone (PCL) fibers. Furthermore, 80% of myosin II inhibited cells (using 50 µM blebbistatin) were oriented within 10° of the average PCL fiber orientation, indicating that cell orientation in response to contact guidance cues in the form of scaffold fibers is not dependent on myosin II [65]. Myosin II inhibition leads to an inhibition of FA maturation. Contact guidance along the fibers can therefore no longer be solely explained by the limited fiber width causing adhesions to mature and grow only in the longitudinal direction. The authors argued that another complementary mechanism is responsible for the aligned adhesions and cells on aligned fibrous substrates. They suggest that the continuous surface of a fiber in the longitudinal direction provides a favorable adhesive substrate to form a sequential series of adhesions, rather than spacing over the gap between several fibers to form new adhesions. This succession in adhesion formation along the direction of the polymer fiber can therefore cause overall cell polarization (Figure 2B) [65].

Further recent research has revealed that a variety of other cellular components and mechanisms also play a role in FA placement and orientation. Swaminathan et al. reported that activated integrins orient and align in FAs as a result of the F-actin retrograde flow. The authors argued that FA elongation originated from the force-dependent directional recruitment of FA proteins and integrins in the direction of the retrograde flow [78]. Huang et al. demonstrated that the stability of the linkage of the FA-adapter protein vinculin to F-actin is direction-dependent. The vinculin-F-actin bond had a 10 times longer lifetime when vinculin was directed towards the pointed (−) end of the actin filament (i.e., the direction of F-actin retrograde flow in a lamellipodium or filopodium) compared to the barbed (+) end [79]. Pontes et al. explored the role of membrane tension on leading edge protrusion and FA placement. They showed that lamellipodium protrusion leaded to an increase in membrane tension, which compressed the lamellipodium [80]. At high membrane tension, the cell front buckles, which leads to the force-dependent maturation of a adhesion row at the base of the buckled region and the formation of new adhesions at the front in a synchronous manner [58,80].

Together, these studies suggest that there is a direction dependency of adhesion protein location and/or orientation in response to anisotropic cues or forces. Anisotropic FA placement and force-dependent FA maturation and stabilization followed by leading edge protrusion in line with the oriented adhesions are suggested to underlie the cellular response towards contact guidance cues smaller than cell size. Several mechanisms for the sensing of contact guidance cues have been put forward, but so far no single, universal mechanism has been demonstrated. In fact, the possibility that a combination of mechanisms induces contact guidance should not be excluded. Furthermore, the geometry sensing mechanisms reported might be cell type or substrate material (or geometry cue type) dependent. While the above-described studies provide fundamental insights into how cells sense and respond to nanoscale contact guidance cues, this knowledge can eventually be of practical relevance for understanding tissue physiology and regeneration. 

## 4. Cell-Scale Geometrical Cues Affect the Cytoskeletal Organization and Nucleus Morphology 

The previous section addresses anisotropic geometrical cues in the nano- or micrometer range, which are primarily relevant for fibrous scaffolds. Many scaffold designs, produced for example using additive manufacturing, also contain features that are of a larger length scale, from tens to hundreds of micrometer. These geometrical features are comparable to or larger than cell size. Cells will recognize the geometry of the struts as substrate curvatures. For this reason, it is of relevance to gain insight into how cells sense and respond to substrate geometries larger than the size of the cell. Before we address this subject in the next section, it is interesting to first focus our attention on the cellular sensing mechanism of 2D geometrical cues equal to cell size. Information obtained from experiments on these 2D subjects can help us better understand the later discussed cellular sensing mechanism on substrate curvatures. These studies on the effect of 2D geometrical cues in the range of the cell size have identified an additional important player in the cellular geometry sensing of cells: the nucleus.

Protein micropatterning on a 2D substrate has been used to systematically study the relationship between extracellular geometric constraints, the organization of various intracellular components, and cell shape. Recent works from our group have shown that micropatterned geometric constraints larger than cell size can strongly affect cell elongation by attenuating the shape fluctuation of cells, highlighting the importance of considering cell mechanics in geometry sensing [81]. At the subcellular level, geometric constraints have further been shown to drastically influence F-actin cytoskeleton organization and nucleus morphology [66,82,83]. Fibroblasts on square-shaped islands (1600 µm^2^) exhibit a mesh-like F-actin cytoskeleton, whereas fibroblasts on rectangular adhesive islands of the same area possess distinct F-actin fibers parallel to the long axis of the cell that bridged over the nucleus. This so-called perinuclear actin cap (or actin cap, in short) also affects the nuclear morphology: the presence of an actin cap, controlled by the cell shape, correlated with a decrease in nuclear height (Figure 2C). The individual actin cap fibers can even indent the apical side of the nucleus, thereby exerting a compressive load on the nucleus [66]. This compressive load of the actin fibers was shown to be actomyosin dependent as nuclear height increased when actomyosin contraction was inhibited [66,84]. It was shown that the nuclear indentations caused by the apical actin stress fibers coincided with an increase of LINC complexes, indicating that the actin stress fibers are firmly anchored at the indentation side (Figure 2D) [67]. 3D reconstruction of confocal images revealed that A-type lamins vertically polarized to a dome structure at the apical side of the nucleus where tension is exerted by the actin cap [85]. The apical stress fibers pressing on the nucleus were also found to affect the internal chromatin organization [66,85,86], histone acetylation levels [82], and telomere dynamics [87]. Telomeres are essential for cellular homeostasis, while histone acetylation is associated with chromatin decondensation and activation of transcriptional activity [82,87]. External geometrical cues at length scales around the cell size can thus affect cell shape, cytoskeletal organization, and nucleus morphology, which in turn can influence intranuclear protein dynamics that play a role in gene expression regulation.

Indentation of the nucleus by actin stress fibers was shown to lead to nuclear confinement accompanied by an increase in intranuclear pressure that can even result in chromatin hernias and nuclear envelope rupture [88]. This shows a similarity to when cells migrate through very narrow constrains that require deformation of the nucleus to fit through [89]. The observation that lamin A increases when actin cap fibers apply a downward push force on the nucleus suggests that the nucleus possesses mechanoresponsive mechanisms to protect its integrity. The effects of the indentation of the nucleus by actin cap stress fibers are reversible when the actin cap disappears. The lamin A network returns to a less compact state and the nucleus returns to a relaxed rounded morphology [67,85]. The changes in chromatin structure initiated by the nuclear compression and indentation are reversible [67,90]. Together, this indicates that the nucleus possesses mechanoresponsive protection mechanisms to prevent excessive nuclear deformation through compressing forces exerted by actin cap stress fibers.

## 5. Substrate Curvature Modulates Cytoskeletal Forces Acting on the Nucleus and Stress Fiber Integrity

Protein micropatterns on 2D substrates have provided valuable insight into the downstream effects of cell shape on intracellular and intranuclear protein organization and dynamics, however such adhesive patterns do not naturally exist in vivo. In the body as well as in tissue engineering scaffolds, cell-scale geometrical cues are primarily represented by 3D curved surfaces such as cavities or cylindrical structures (e.g., in the form of scaffold struts, natural vessels, or large collagen bundles) [91]. Substrate curvatures with radii of curvature similar or bigger than cell size have been shown to remarkably affect the morphology of human Bone Marrow Stromal Cells (hBMSCs) [68]. Concave (valley shaped) and convex (hill shaped) substrates can induce radically different attachment morphologies. hBMSCs on concave hemispherical substrates (diameter = 250–750 µm) stretch upward, minimizing the contact area of the cell to the substrate with only district attachment points at the cell periphery. On convex hemispherical substrates, the cell body remains in full contact with the substrate and the nucleus is flattened and sometimes even presents a bent (bean-like) shape. Actin cap stress fibers indent on the nuclear membrane and lamin A levels are elevated on convex substrates compared to on flat and concave substrates, indicative of a compressive force on the nucleus (Figure 2E) [68]. Pieuchot et al. used a sinusoidal hills and valley array substrate (sinus wave length of 100 µm, maximum amplitude of either 5 or 10 µm) and demonstrated that migrating mesenchymal stem cells (MSCs) avoid convex regions. The nucleus seems to be the leading factor in this behavior. Nuclei in migrating cells are translocated from the concave region to concave region, remaining excluded from convex regions. Nuclei on concave surfaces are more spherical compared to on flat regions, suggesting that the nucleus is in a mechanically relaxed state when positioned in concave regions [92]. Computational modeling gives us further insight into the effect of substrate curvature on the mechanical state of adherent cells. In line with the abovementioned experimental studies, the computational model presented by Vassaux and Milan predicts an increasing vertical compressive pressure on the nucleus when the degree of convexity of the substrate increases (radii of curvature ranging from 75–500 µm). Stress fibers function in a bent configuration on convex surfaces, resulting in a vertical force pointing downward that applies a pushing force on the nucleus. On the other hand, on concave surfaces, the stress fibers were almost straight in the horizontal plane, resulting in a reduction in vertical straining forces and a rounder nucleus [93].

Beside the compressive force that stress fibers can transmit onto the nucleus, the integrity of the stress fibers itself is also influenced by curved substrates. A pioneering study by Dunn and Heath demonstrated that substrate curvature-induced stress-fiber bending impairs the coherence of the actin cytoskeleton and may even lead to actin fiber fragmentation [94]. Computational models also showed that it is energetically favorable for a stress fiber to be in a straight configuration [95]. Bending of stress fibers would impede the contractility of the cell [96]. Biton and Safran presented a theoretical model that showed that the cellular response towards a cylindrical substrate depends on the competition between the stress fiber bending energy and the deformation energy [95]. In cells with developed and aligned stress fibers, the stress fiber bending energy dominates and cells align along the cylinder axis to prevent stress fiber bending and lower the overall elastic energy. On the contrary, in cells with thinner stress fibers, the active contractility of the cell is the dominant factor determining the cellular orientation. These cells are predicted to orient perpendicular to the cylinder axis because the deformation due to the substrate curvature partly cancels out the active contractile deformation, lowering the overall elastic energy [95]. Recently, Bade et al. identified that actin stress fibers below and above the nucleus respond differently in mouse embryonic fibroblasts and human vascular smooth muscle cells cultured on cylinders [97]. Actin cap stress fibers, running over the nucleus, align along the longitudinal axis of the cylinder, while the shorter basal stress fibers, below the nucleus, orient in the circumferential direction. Rho activation (which regulates smooth muscle contraction) lead to disassembly of the actin cap stress fibers, whereas basal stress fibers became thicker and strongly align in the circumferential direction of the cylinder [97]. The authors speculate that the activation of Rho leads to a contractility-dominated regime that can override the bending-dominated regime [97]. We recently directly compared the actomyosin cytoskeleton in hBMSCs on convex spherical and cylindrical substrates [69,98]. On spherical surfaces, cells adapt their morphology to the curved substrate, while on cylindrical substrates cells elongate and direct their cell body towards the longitudinal direction of the cylinder, thereby largely avoiding the curvature of the substrate (Figure 2F). More pronounced stress fibers and a higher F-actin signal intensity per cell was found on cylindrical substrates in comparison to spherical substrates. Interestingly, higher levels of myosin light chain phosphorylation, indicative of non-muscle cell contractility, were found in cells bent on spherical surfaces compared to cells on cylinders [69]. These observations suggest that hBMSCs have pronounced F-actin fibers in an unbent situation. The lower F-actin levels and higher phosphorylated myosin levels observed on spherical substrates suggest that more myosin II has to bind to the actin cytoskeleton to balance the reduction in F-actin stress fibers to maintain cytoskeletal tension in cells in a curved configuration. 

The above mentioned studies suggest a delicate interaction between cellular bending, stress fiber organization, and compressive forces on the nucleus that together orchestrate the cellular response towards curved substrates (Figure 3). It is of interest to directly compare various cell types with differently organized cytoskeletons on anisotropic curved substrates (e.g., cylinders) in the future. In particular, a comparison of cells that naturally possess pronounced apical stress fibers (such as fibroblasts and mesenchymal stromal cells) with cell types that do not have aligned apical stress fibers or possess a cortical cytoskeleton could advance our knowledge on the role of the cytoskeleton on the cellular orientation response on strut-like structures.

## 6. Discussion and Future Perspectives

### 6.1. Identifying the Mechanical Roles of the Subcellular Elements in Cellular Geometry Sensing

Curvature guidance seems to result from a mechanosensing and mechanoresponse mechanism. Since curvature sensing is relevant in the context of biomaterial-guided regeneration, it is critical to identify the precise mechanisms underlying the curvature guided cellular response and its effect on subsequent tissue organization. There is evidence from several studies discussed in this review that, when a cell with apical stress fibers is subjected to a curved substrate (e.g., a scaffold strut), outside-in signaling takes place, resulting in nucleus deformation due to indenting stress fibers and/or impeded stress fiber integrity. An inside-out signaling follows that is aimed to regain nucleus relaxation (e.g., by lifting the cell body upwards towards a less bent configuration in concave pits) and/or restore stress fiber stability (e.g., by reorienting towards a less bent configuration). If this is not possible (e.g., on convex spherical substrates where cells have no possibility to avoid the substrate curvature by rearranging their morphology or orientation), the cells cope by applying biochemical changes in the actomyosin cytoskeleton to maintain cytoskeletal integrity in a bent configuration. Further studies that closely monitor (or model) the nuclear morphology and stress fiber organization in several cell types in response to a variety of curved surfaces will be necessary to better understand the biomechanical mechanisms behind this curvature guidance response. This information in turn is relevant for selecting strut sizes and geometries in scaffold designs. The observation that cell bending induces nuclear indentation by the apical stress fibers [68], while other studies report that cell bending impedes apical stress fiber integrity [94], may seem contradictory. This motivates studies examining the forces (magnitude and direction) stress fibers can exert in bent and straight configurations. It further highlights the need to study both the nuclear and stress fiber response simultaneously and monitor the timescale on which adaptations in nucleus morphology and the cytoskeletal organization and orientation take place. 

### 6.2. Contribution of Cytoskeletal Subtypes in the Cellular Response Towards Geometric Cues

Current studies mainly focused on the organization of the actin cytoskeleton when investigating the contact and curvature guidance response. Actin fibers interact with the two other cytoskeletal components, microtubules and intermediate filaments [99]. Microtubules and intermediate filaments are, like actin fibers, also coupled to the nucleoskeleton through LINC complexes [100]. The cytoskeletal subtypes are often studied separately, however, there is growing evidence that the interconnections between the cytoskeletal fibers result in coupled functions that control cell morphology, polarization, and migration [99,101,102,103,104]. In other words, the overall functionality of the cytoskeleton cannot be simply represented by the accumulation of separate functions of the three cytoskeletal subtypes [99]. This makes it difficult to study the contribution of a separate cytoskeletal subtype to a specific function in the cell. A future challenge lies in further unraveling the separate and additive functions of the cytoskeletal subtypes to deepen the mechanistic understanding of the cytoskeleton in general and the role of the cytoskeletal subtypes in mechanosensing and mechanoresponse behavior.

### 6.3. Deeper Insight into the Role of the Nucleus in Cellular Geometry Sensing 

To specifically determine the role of the nucleus in the cellular orientation response towards geometric cues, it is of interest to study cells with divergent nuclear properties. Lamin deficient cells have a softer nucleus and do not have an actin cap, while lamin B1 overexpression can lead to a stiffer nucleus [105,106,107]. Cancer cells are known to have highly deformable nuclei and are associated with aberrant nucleus shapes [108,109]. Enucleated cells, cells where the nucleus was completely removed, have previously been shown to have unaffected migratory behavior on 1D (in the form of a microprinted protein line) and 2D migration behavior, but showed slower migration behavior in 3D collagen matrices in comparison to untreated cells [110]. Together, these cells form interesting model cell types to study the role of the presence of a nucleus and nuclear mechanics on the response of cells towards geometrical cues. However, it should be noted that the contribution of the nucleus and the actin cap are difficult to study individually, because of their interconnection. Depletion of lamin or removal of the entire nucleus will therefore also affect the presence of the actin cap fibers. Besides the effect on the nuclear lamina, chromatin, and telomere organization, it is also of interest to broaden our knowledge towards the effect of geometrical cues on other nuclear components. 

### 6.4. From 2D and 2.5D towards a 3D Configuration

Actin cap fibers only exist in a situation where a cell has a clear apical and basal side (e.g., on a 2D substrate but also when a cell is attached on one side to struts of a 3D porous scaffolds, while no contact to other struts or features is made on the apical side of the cell). This is as opposed to a 3D environment where a cell is completely surrounded by the extracellular matrix and where no apical or basal side can be appointed. In vivo, contact guidance and curvature guidance cues can for instance be present in the form of aligned collagen fibers, vessels, or myofibers [111,112]. These structures are surrounded by other tissues and extracellular matrix. Upon implantation of a cell-free scaffold in a tissue engineering approach, infiltrating cells can initially perceive the scaffold struts as a locally curved substrate (depending on the internal architecture and distance between the scaffold struts). However, at a later time point, when cells have started to produce ECM and the scaffold has started to degrade, the cells can perceive their surroundings as a 3D environment. This has motivated studies to move towards a 3D approach, by studying contact guidance in isotropic and anisotropic collagen matrices [113]. It also raises the question to what extent our knowledge from in vitro studies on 2D contact guidance and 2.5D curvature guidance can be extended towards an in vivo situation when cells are fully embedded in a 3D matrix. Microcontact printed lines on a 2D substrate or hemispherical and cylindrical 2.5D substrates are used as study platforms that mimic an in vivo situation. An important advantage of using 2D and 2.5D study platforms is that they allow detailed microscopic examination of intracellular structural components of the cell and the control to specifically subject the cell to highly controllable cues. This has revealed important insight into the recognition and response mechanism of cells towards geometrical cues. A future step could involve an in vitro study platform where a clear distinction between the apical and basal sides of the cell is absent while maintaining the ability for high-resolution imaging and to systematically study the cellular response towards specific cues.

### 6.5. Cellular Response towards Multiscale Geometrical Cues 

While studies presenting cells with one geometrical cue at a time have provided valuable insights, in vivo cells are often simultaneously subjected to multiple geometrical and mechanical cues. Experimental platforms are emerging that subject cells to a more complex environment with hierarchical multiscale cues. It was for instance shown in a 2D setting that mesoscale cues (> 100 µm) in combination with microscale cues (1–2 µm) act synergistically to enhance the alignment of neo-tissue formation by MSCs when these cues are in the same direction. However, the mesoscale confinement cues overrule the microscale cues when presented perpendicular to each other [114]. In a migration study of breast epithelial cells, it was demonstrated that while nanoscale cues (ridges, 800 nm ride and groove width, 600 nm height) promote migration in comparison to a flat surface, the microscale constraints (collagen-coated pattern with 30–120 µm widths) are the dominant factor in controlling the migration distance [115]. Recently, we demonstrated that curvature guidance cues in the form of cylindrical substrates can overrule nanoscale contact guidance cues in the form of a dense anisotropic network of collagen fibrils aligned perpendicular to the cylinder axis [69]. Together, these studies hint that multiple geometrical cues can enhance or compete with each other in a complex manner that we do not fully understand yet. A systematic exploration of how cells perceive and process multiple geometrical cues, also in the presence of different spatial patterns of cell-binding ligands, is thus of interest to better understand the cell’s decision-making in complex multiscale environments. Furthermore, substrate materials that allow dynamic alterations of their properties (and thereby dynamic control of cues presented to the cell) in combination with high resolution live imaging represents another interesting path to study the immediate effect of cues on the intracellular structural organization and response.

## 7. Conclusions

Lately, it has become clear that cells can sense and respond to geometrical cues over a wide range of length scales, from the nanometer to millimeter scale. Moreover, advanced scaffold production techniques allow the controlled and reproducible production of complex scaffold architectures. This review has discussed the state-of-the-art knowledge on how cells sense geometrical cues of different size ranges. The challenge is to translate the knowledge gained from 2D and 2.5D model systems towards an informed 3D scaffold design that has the potential to guide cell and tissue organization in a predetermined direction. In order to reach this goal, a concerted, systematic effort is necessary to study the cellular sensory and responsive mechanisms towards a variety of geometries and materials with highly defined properties. Moreover, using different cell types and cells with modified nuclear or cytoskeletal properties on this library of different geometries and materials in combination with high-resolution imaging could further improve our knowledge on the role of different intracellular components on the overall cellular response. In combination with computational modeling, this has the potential to further advance our knowledge on the sensory mechanism of cells towards geometrical cues of different size and origin. This knowledge can, in turn, form the basis of scaffold design principles for specialized tissue engineering approaches. Customized scaffold designs could, for instance, promote the formation of a native-like tissue organization in tissue-engineered constructs. Interdisciplinary exchange of expertise, materials (e.g., polymers, but also scaffolds containing highly defined geometries), detailed standardized operation procedures, and experimental results are essential for the success of this approach.

## Figures and Tables

**Figure 1 materials-13-00963-f001:**
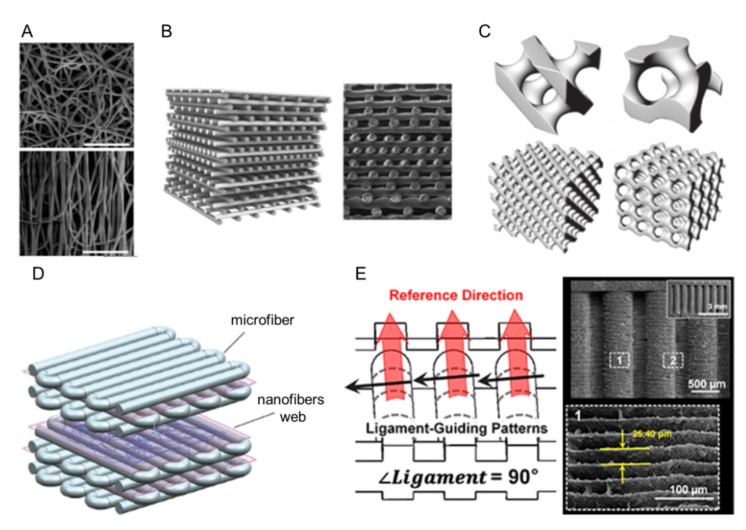
Examples of scaffold designs. (**A**) Electrospun polycaprolactone (PCL)-bisurea scaffolds with an isotropic (top) or an anisotropic organization (bottom). Fiber diameter is approximately 6 µm. Scale bar = 100 µm. Image adapted from [45]. (**B**) Computer-aided design (CAD) 3D model (left) and scanning electron microscopy (SEM) image of a scaffold produced by 3D plotting from a blend of corn starch with PCL (30/70 wt %). The fiber thickness (i.e., layer thickness) was 0.19 mm. A gradient in fiber distance was applied in which the fiber distance progressively decreased by 0.05 mm, resulting in a fiber distance of 0.75 mm in the outer layers and a fiber distance of 0.1 mm in the center of the scaffold. Image adapted from [46] with permission from Elsevier. Note the difference in fiber size between (**A**) and (**B**). The fiber shape is similar. However, the fiber diameter in (**A**) is much smaller than the size of a cell, while the fiber diameter in (**B**) is similar or larger than the cell size. (**C**) Scaffold designs based on triply periodic minimal surfaces (TPMS) have highly defined internal geometries. A diamond structure is depicted on the left and a gyroid structure on the right (top images show one unit cell of the structure, bottom images the CAD design of a scaffold with 4 × 4 unit cells). The Gaussian curvature distribution of the scaffold struts can be calculated and by adjusting the number of unit cells and scaffold dimensions a scaffold can be designed with predefined Gaussian curvature distributions. Image adapted from [28]. (**D**) The CAD model of a hierarchical scaffold that combines gel microfibers (20 wt % gelatin solution and 4 wt % sodium alginate, diameter = 440 µm) produced by 3D bioprinting and PCL nanofibers (diameter = 190 nm) produced by electrospinning. Image adapted from [33]. (**E**) Hierarchical PCL scaffold with cylindrical structures (0.5 mm diameter) that contain microgroove patterns aligned perpendicular to the longitudinal reference direction of the cylindrical structures (thickness = 25.4 µm). Microgroove patters aligned along the longitudinal direction of the cylinders (0°) and at a tilted 45° angle were also produced. Image adapted from [20].

**Figure 2 materials-13-00963-f002:**
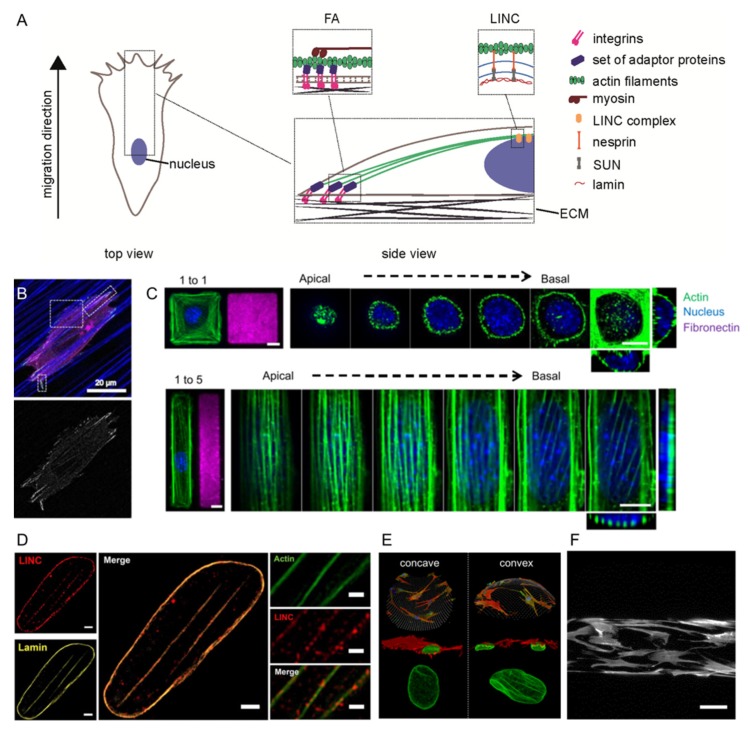
Cells can sense the physical properties of the environment. (**A**) Schematic representation of the connection from the ECM to the nucleus via focal adhesions (FAs), actin filaments, and linker of the nucleoskeleton with the cytoskeleton (LINC) complexes. Note that many more intracellular components play a role in cell–matrix adhesions and the transmission of extracellular signals towards the inside of the cell. This figure summarizes the intracellular components that are discussed in this review. (**B**) HT-1080 cells expressing Ruby-Lifeact indicating actin in magenta (top) and EGFP-paxillin indicating paxillin in FAs (bottom) on 2D polycaprolactone scaffolds with aligned fibers (blue, fiber diameter 700 nm). Adhesions were primarily formed along the aligned fibers. Image reused from [65]. (**C**) Micropatterned geometric constraints comparable to cell size (island area = 1600 µm^2^) affect cell shape, F-actin cytoskeleton organization, and nucleus morphology. On square fibronectin islands, fibroblasts exhibit a mesh-like F-actin cytoskeleton, whereas on rectangular islands have F-actin fibers that run parallel over the nucleus. Note the difference in nuclear height between the cells on square and rectangular islands (images on the far right). Scale bars represent 10 µm. Image reused from [66] with permission from Elsevier. (**D**) The actin cap fibers indent on the nucleus in human umbilical vein endothelial cells (HUVECs). An increase in LINC complexes and lamin at the location of the actin fiber indentation sites indicate that the actin fibers are anchored to the nucleus and apply a compressive force on the nucleus. Scale bars represent 1 µm. Image reused from [67]. (**E**) Curved substrates can affect cellular and nuclear morphology. Concave spherical surfaces enabled the cell body of human bone marrow stromal cells to lift off the surface, on convex spherical surfaces cytoskeletal tension induced nuclear compression and deformation (F-actin in red, lamin A in green). Image reused from [68]. (**F**) Human bone marrow stromal cells align along the longitudinal axis of a convex cylindrical substrate (F-actin in grey, cylinder diameter = 250 µm). Scale bar = 100 µm. Image adapted from [69].

**Figure 3 materials-13-00963-f003:**
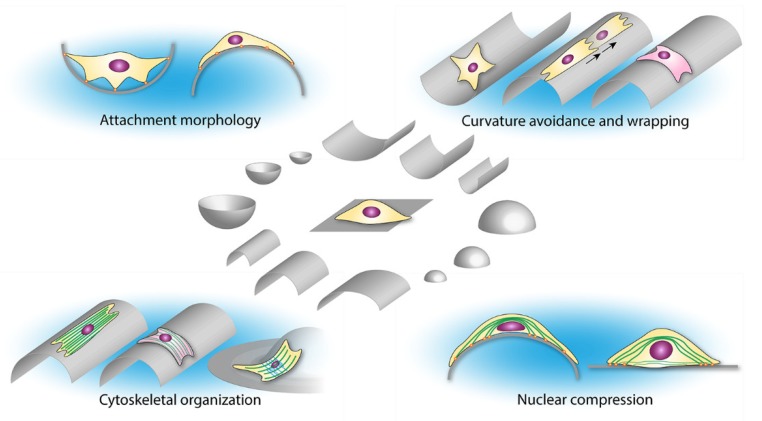
Reported observations and proposed mechanisms of cellular geometry sensing at different length scales.

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
