# Peer review of "Cellular Geometry Sensing at Different Length Scales and its Implications for Scaffold Design"

_materials, 2020, doi:10.3390/ma13040963_

Round 1

Reviewer 1 Report

This review focuses on how cells sense geometrical cues of different sizes and origins; the mechanisms that cells employ to sense geometrical guidance cues smallerthan cell size; the sensing of substrate curvature. F-actin cytoskeleton organization and nucleus morphology are shown to be significantly involved in cellular geometry sensing of cells.

I think it can be published as it is.

I have more comments to this paper:  

Different types of cells respond differently to the same cell scaffold. For example, neuron cells are very sensitive to the fiber directions of the scaffold. Could the authors address more on the issue of how scaffolds match to the different cell types? Especially for the neuron regeneration issues.

We have some suggestions for the English corrections for your reference. 

Line No.

Words to be corrected

Correction

11

on

of

12

is

are

20

on

of

21

Cell and tissue architecture

the architecture of cells and tissues/

cellular and tissue architecture

22

on

of

30

function

functions

36

cell

cellular

37

to

to the

38

re-arranging

rearranging

38

on

of

39

influences

influences the cellular

39

To better understand

to gain a better understanding of

40

but

and/

as well as the

41

the substrate

substrate

42

direct

direct the

55

and

and the

80

on

of

82

a

the

143

they

but they

150

on

to

152

cytoskeleton

the cytoskeleton

166

sections

sections,

201

can for instance be

can, for instance, be

221

has a stiffness

has stiffness

228

of

of the

252

direction dependent

direction-dependent

268

above described

above-described

277

in

into

312

requires

require

322

in

into

332

substrates

substrates,

342

in

into

420

however

however,

429

in

into

448

by

by the

463

in

into

466

high resolution

high-resolution

478

collagen coated

collagen-coated

492

controlled

the controlled

500

high resolution

high-resolution

503

can in turn

can, in turn,

504

could

could,

505

for instance

for instance,

505

tissue engineered

tissue-engineered

Author Response

We thank the reviewer for the suggestion. In this manuscript, we discuss geometry sensing from the perspective of the cellular machineries that enable such sensing. These machineries (e.g., focal adhesion, cytoskeleton, nuclear mechanics) are generic and varying across various adherent cell types, including neurons. While an in-depth discussion of neuron geometry sensing per se is not our aim, we agree that neuron regeneration using scaffolds presents one of the potential applications and can be used as an example to establish the relevance of this research line. As such, we have added several examples of the in vivo relevance of cellular geometry sensing, including for neurons, in section 1 (line 78–85) of the revised manuscript.

We have also carefully checked the entire manuscript again and corrected any linguistic error found.

Reviewer 2 Report

The manuscript titled ‘Cellular geometry sensing at different length scales and its implications for scaffold design’ is a short, concise review of the recent progress across several fields on how cells sense geometrical cues. It is very well written and clear, and in particular the last critical future looking overview in a good addition to the field. I would have liked to see the following points addressed in the manuscript for it to be more accessible to a wide audience and offer additional critical thought.

I found Figure 2A a useful schematic overview of FA and LINC for a general audience. I think the manuscript would really benefit from another schematic overview summarising different sensing mechanisms/processes/observations at different scales. While I appreciate the mechanisms are not always elucidated, even a schematic summary of common observations would be useful (eg. effects on FAs, cytoskeleton, nuclei under different conditions) Again, while Figure 2 provides useful examples of the actin skeleton chnages, there are no examples of changes to focal adhesions, and a whole section of the manuscript is dedicated to this The authors discuss the pros and cons of using highly simplistic systems such as 2D substrates with grooves in answering mechanistic questions, which is good to see. I would also like to see a comment about any similar observations of processes, cell changes, or mechanisms that have been observed in more complex systems such as tissue engineered constructs or even native tissue. If such examples do not exist in any field, a comment about this would be useful. A somewhat more detailed discussion of the effects of multiple factors would be beneficial. In particular I would have liked to see some discussion about specific cell binding sequences that were provided in different systems and how these correlated with observed changes. The authors discussed the fact that it is never a single factor causing observed changes- any suggestions or examples of how to study these complex interactions between different construct properties in a systematic manner? Finally, the title has the words ‘implications for scaffold design’- I would  like to see at least one example where this was put to practice- is there an example of rational scaffold design where it is clearly demonstrated that the changes/improvements in design affected performance?

Author Response

Reviewer’s comment:

The manuscript titled ‘Cellular geometry sensing at different length scales and its implications for scaffold design’ is a short, concise review of the recent progress across several fields on how cells sense geometrical cues. It is very well written and clear, and in particular the last critical future looking overview in a good addition to the field. I would have liked to see the following points addressed in the manuscript for it to be more accessible to a wide audience and offer additional critical thought.

Authors’ response:

We are pleased that the relevance and contribution of our article to the field can be readily appreciated. We have addressed all remarks, as detailed below.

Reviewer’s comment:

I found Figure 2A a useful schematic overview of FA and LINC for a general audience. I think the manuscript would really benefit from another schematic overview summarising different sensing mechanisms/processes/observations at different scales. While I appreciate the mechanisms are not always elucidated, even a schematic summary of common observations would be useful (eg. effects on FAs, cytoskeleton, nuclei under different conditions) Again, while Figure 2 provides useful examples of the actin skeleton chnages, there are no examples of changes to focal adhesions, and a whole section of the manuscript is dedicated to this

Authors’ response:

We thank the reviewer for the excellent suggestion. We have added a new figure (Figure 3) containing a schematic illustration of several reported observations and proposed mechanisms of cellular geometry sensing.

Reviewer’s comment:

The authors discuss the pros and cons of using highly simplistic systems such as 2D substrates with grooves in answering mechanistic questions, which is good to see. I would also like to see a comment about any similar observations of processes, cell changes, or mechanisms that have been observed in more complex systems such as tissue engineered constructs or even native tissue. If such examples do not exist in any field, a comment about this would be useful.

Authors’ response:

We have added a few examples of the in vivo relevance of cellular geometry sensing in section 1 (line 78–85) of the revised manuscript.

Reviewer’s comment:

A somewhat more detailed discussion of the effects of multiple factors would be beneficial. In particular I would have liked to see some discussion about specific cell binding sequences that were provided in different systems and how these correlated with observed changes. The authors discussed the fact that it is never a single factor causing observed changes- any suggestions or examples of how to study these complex interactions between different construct properties in a systematic manner?

Authors’ response:

We agree with the reviewer that it is indeed important to gain better understanding of the effect of multiple cues. In this manuscript, we focus on geometrical cues, and section 6.5 is dedicated to exploring cellular response to multiple, and sometimes competing, geometrical cues, including several examples. The reviewer raised a point on specific cell binding sequences that is outside the scope of our present manuscript. Nevertheless, we have added a short note emphasizing the need to explore this further in section 6.5 (line 496–498) of the revised manuscript.

Reviewer’s comment:

Finally, the title has the words ‘implications for scaffold design’- I would  like to see at least one example where this was put to practice- is there an example of rational scaffold design where it is clearly demonstrated that the changes/improvements in design affected performance?

Authors’ response:

The crucial influence of geometry has only started to be recognized recently, and a rational scaffold design based on cellular geometry sensing is not yet a commonly taken approach nowadays, mainly for two reasons. First, there is still lack of fundamental knowledge of how cells sense and respond to geometrical cues in the scaffold—the present review manuscript is in fact an attempt to consolidate the current knowledge. Second, multiple geometrical parameters are often interlinked in scaffold due to limitations in the fabrication method. This is why understanding cellular response towards multiscale geometrical cues is critical (as already discussed in section 6.5 of the manuscript). For these reasons, we believe that this is a very promising avenue and we eagerly anticipate research outcomes in this direction.

To further highlight this point, we have revised our text in section 1 (line 64–68) of the manuscript, including two recently published review articles that specifically address the tuning of scaffold structural parameters in directing cellular response (Jenkins & Little, npj Regen Med 2019, 4:15; Ameer et al, J Funct Biomater 10:30, 2019).